# Studying Behaviour Change Mechanisms under Complexity

**DOI:** 10.3390/bs11050077

**Published:** 2021-05-14

**Authors:** Matti T. J. Heino, Keegan Knittle, Chris Noone, Fred Hasselman, Nelli Hankonen

**Affiliations:** 1Faculty of Social Sciences, University of Helsinki, P.O. Box 54, 00014 Helsinki, Finland; matti.tj.heino@helsinki.fi (M.T.J.H.); keegan.knittle@helsinki.fi (K.K.); 2School of Psychology, National University of Ireland, H91 TK33 Galway, Ireland; chris.noone@nuigalway.ie; 3Behavioural Science Institute, Radboud University Nijmegen, Postbus 9104, 500 HE Nijmegen, The Netherlands; f.hasselman@pwo.ru.nl

**Keywords:** complex systems, wellbeing, methodology, behaviour change

## Abstract

Understanding the mechanisms underlying the effects of behaviour change interventions is vital for accumulating valid scientific evidence, and useful to informing practice and policy-making across multiple domains. Traditional approaches to such evaluations have applied study designs and statistical models, which implicitly assume that change is linear, constant and caused by independent influences on behaviour (such as behaviour change techniques). This article illustrates limitations of these standard tools, and considers the benefits of adopting a complex adaptive systems approach to behaviour change research. It (1) outlines the complexity of behaviours and behaviour change interventions; (2) introduces readers to some key features of complex systems and how these relate to human behaviour change; and (3) provides suggestions for how researchers can better account for implications of complexity in analysing change mechanisms. We focus on three common features of complex systems (i.e., interconnectedness, non-ergodicity and non-linearity), and introduce Recurrence Analysis, a method for non-linear time series analysis which is able to quantify complex dynamics. The supplemental website provides exemplifying code and data for practical analysis applications. The complex adaptive systems approach can complement traditional investigations by opening up novel avenues for understanding and theorising about the dynamics of behaviour change.

## 1. Introduction

Behavioural interventions often fail to produce sustainable effects [1], especially when transferred from one context to another. One core interest of behaviour change science then, is to improve our understanding of interventions’ mechanisms of action. Behavioural theories identify hundreds of potential “determinants” of behaviour, that is, factors that potentially influence the behaviour of interest. These determinants constitute the channels through which behaviour change techniques might influence behaviour [2]. Determinants range from social cognitions such as self-efficacy and attitudes, to biological factors, and elements of the social and built environments in which behaviours take place [3]. When studied using typical factorial designs and linear statistical models, the relationships between causal precedents and behaviour change are assumed additive, constant and linear (i.e., the outputs are proportional to the inputs). However, it is our position that this offers researchers and the public an inaccurate and imprecise understanding of behaviour change. We should instead consider the relevant factors as complex, potentially non-linear, and dynamic.

The evaluation of behaviour change interventions often involves randomly assigning participants to receive an intervention of interest or a comparator, and measuring subjective and objective indicators of behaviour [4]. Usually, these measurements occur immediately before and after the delivery of the intervention, though sometimes additional follow-up measurements may take place weeks or months later. This is the classic Randomised Controlled Trial (RCT) design and the data from such studies are most often analysed using statistical techniques that are specific cases of the General Linear Model. In this paper, we refer to this as the “conventional approach.” To assess whether an intervention was more effective, on average, than a comparator, comparing averages in RCTs can be purposeful and acceptable. This method can answer questions such as “Does the intervention influence the target behaviour?”, and “Do cohorts differ from each other?” However, using so few measurement points to study behaviour change mechanisms (“How do intervention participants change?”) may present problems.

Currently, mechanisms of change within behaviour change interventions are typically studied using mediation analysis [5], where the impact of X (e.g., an intervention) on Y (e.g., a behavioural outcome) is modelled to pass through a third variable M (e.g., a theoretical determinant of the behavioural outcome targeted by the intervention). In the presence of classical mediation, the path X-Y would be reduced to near zero when adding M to the model. If this were observed, the researcher would conclude that there is evidence for mediation. In the case of behaviour change interventions then, one would conclude that the intervention (X) changed behaviour (Y) because it changed important theoretical determinants of the behaviour (M). For a case example from members of the current author group see [6].

Inferring mechanisms from contemporary mediation analyses is problematic on various grounds [7,8,9,10,11,12]. Of particular importance to human behaviour change, however, is that the accuracy of mediation analysis depends on four key assumptions [13]: (1) The number of variables involved is small, and dynamics can be meaningfully assessed with only a few time points; (2) The process of change is the same for all individuals, e.g., follows the same sequence; (3) The dynamic between variables is linear, additive, and does not change in time; and (4) The included variables are not entangled with the context, omitted variables, or each other in bi-directional recursive relationships. Researchers can, of course, include more variables (leading to new issues, e.g., mixing up mediators, confounders and colliders [14,15] or lowering the already worrisome statistical power [16]), try to incorporate non-linear effects [17], and add more time points (for caveats regarding latent variable modelling, see, e.g., [18,19]). However, limiting the notion of a mechanism to a (multiple) mediation/moderation problem narrows our understanding of how changes occur over time [20,21].

During the first two decades of the 21st century, behaviour change researchers began extending the traditional approach and embracing designs with an increased focus on temporal processes [22,23]. Recently, alternative solutions stemming from complex systems science [24,25] have become increasingly accessible and helpful in understanding change processes. We will explore these ideas and how they help us surpass traditional assumptions. In what follows, this paper will (1) outline the complexity of behaviours and behaviour change interventions; (2) introduce readers to some key features of complex systems and how these can be applied to human behaviour; and (3) provide concrete suggestions for how researchers can better account for the implications of complexity in analysing behaviour change mechanisms.

### 1.1. What Are Complex Systems?

A system is “a delineated part of the universe which is distinguished from the rest by an imaginary boundary” [26], although other definitions exist [27]. Many things—an airplane, a car, a robot, a central nervous system, a school, a community, a society—can be conceptualised as systems. This paper focuses on individual people, which are complex systems. Complex systems can be characterised as webs of many *interdependent self-organising* parts that operate without central control, whose interactions give rise to *emergent* properties and behaviours, *irreducible* to a sum of the parts [28]. The future behaviour of a complex system strongly depends on its *unique history of interactions*, that is, past experience. Additionally, the system adapts to its environment and actors therein, *coevolving* with each other to create macro-level behaviour. This dynamic is difficult to predict and is usually not changeable in a stepwise engineering sense [29]. The italicized characteristics above distinguish complex systems from those which are just complicated. Highly complicated systems (e.g., an airplane), unlike complex ones (e.g., an organism), cannot self-organise to function adaptively when one part is removed [30]. Guides to basic terminology of complexity for scientists working with health behaviours can be found in [30] as well as Table 1 of [29].

### 1.2. The Relevance of Complexity for Behaviour Change

The promise of complex adaptive systems approaches in health behaviour change research has been previously discussed by, e.g., [31], but over a decade ago, not many empirical solutions were easily accessible to investigators in this field. Recently, methods to study complex adaptive systems in health behaviour change have been presented [32], with a focus on simulation and qualitative methods. This article discusses novel quantitative solutions, which have recently become available, to investigate behaviour change phenomena with a complex systems lens. 

To paint a picture of just how complex the behavioural world is, take the case of physical activity. Already three and a half decades ago, more than 30 influences on (or “determinants of”) physical activity were being considered, along with calls for better understanding of their dynamics, interactions, and the time scales over which these develop [33]. While any influence (e.g., intention, attitude) could have a direct relationship with physical activity, some rely on interactions with other influences to affect behaviour [34,35]. Furthermore, these interactions may be moderated by additional factors, and by other variables which themselves have no direct relationship with physical activity, with synergistic and opposing effects which may themselves depend on whether some threshold is exceeded. The extent to which all known (and unknown) influences on physical activity interact with one another presents a map of practically infinite, intertwined ‘routes’ to initiating and maintaining physical activity.

The role of time brings added complexity to this behavioural world, as dynamic patterns change over time and at varying frequencies [36,37]. For example, fluctuations in physical activity clearly occur within a day, as most individuals are (at least in the absence of highly sedentary working conditions and considerable somnambulism) more active while awake than while asleep. Fluctuation also occurs over the course of a week, as activity levels tend to be higher on weekdays than on weekends [38]; over the course of months, as activity levels are higher in warmer seasons and lower in colder ones [39] and over the course of years, as activity levels tend to decline with age [40]. How determinants—which are postulated to comprise the mechanisms underlying changes in behaviour—fluctuate and interact with the fluctuations in behaviour, is largely unknown.

Human behaviour is complex, and while we have formulated theoretical constructs to be as amenable as possible to linear methods of analysis, this may obscure important characteristics of behaviour change. Why are linear models inappropriate for many of our research questions in the behavioural sciences? First, with many non-linear interactions across time scales, mechanistic causality (including mediation and moderation) becomes suspect or intractable [19,41,42]. Second, traditional statistical analyses start from the simplification that everything is independent from everything else. In actuality, nearly everything eventually depends on everything else, contributing to what Paul E. Meehl [43] seminally coined as “the crud factor”. Jacob Cohen, the developer of power analysis, similarly exclaimed that (in the absence of randomisation), the nil hypothesis of no effect is always a priori false [44]. These well-known ideas demonstrate violations of the classical assumptions regarding independence and interference [45,46]. In the same vein, forecasting in complex systems is notoriously difficult [47,48,49], making hypothesis testing—which is, after all, the test of a prediction—in intervention evaluation, a curious challenge. 

Complexity science starts from the assumption that everything is intertwined, and can provide us with new hypotheses that respect the complexity of the phenomena under study [50]. This is necessary, because a conventional linear analysis will only yield accurate results when underlying assumptions are met: that the components in the model are independent, with additive effects that can be decomposed and attributed to their causes (e.g., beta coefficients in multiple regression). If, on the other hand, these “component-dominant” dynamics are not driving the system, and instead, the effects are intertwined, overlapping and inseparable [51], the dynamics are “interaction-dominant”, and the replication and generalisation issues for results stemming from the linear analysis are almost inevitable [46]. Instead, intensive longitudinal methods are necessary to monitor how processes unfold. This information can then be used to dynamically tune interventions in real time, making success less dependent on having a correct program theory at the outset [52,53]. Made possible by N-of-1 methodologies [54], this goal has been recently pursued by using control systems engineering approaches [55] and just-in-time adaptive interventions [56], among others.

Although behaviour change maintenance has been theorised at length [1], existing theories do not incorporate complex systems principles, which might overcome the aforementioned issues. From the viewpoint of complex systems science, the effects of behaviour change interventions can be considered as shocks to the system in which they take place. The aim of the intervention shock is to alter the system’s status, pushing against existing forces to affect change [57,58]. This is akin to attempts to work against gravity, which pulls a ball to the bottom of a valley. In this example, the valley represents a relatively stable state, also known as an attractor (see Figure 1) [59,60]. Taking the analogy further, pushing the ball outside of the valley may lead it to roll off a peak, ending up in a deeper valley than where it started (i.e., a more stable, deep-rooted state). A complex systems perspective implies that, even in the event of a successful intervention, stabilizing a system in a more functional state may require at least as many resources as the initial change itself [61]. In general, while complex systems may often be impossible to control precisely, they can be stewarded approximately, while allowing for variability stemming from self-organisation to flourish instead of trying to iron it out [62,63]. The necessity of a complex systems approach is increasingly recognized, including within the UK Medical Research Council’s recently updated guidance for developing and evaluating complex interventions [64].

Having now undergone a brief conceptual introduction to complexity, we can describe behaviour change as “a collection of contextualised processes that are nontrivially specific to each individual, and which form a complex interconnected system that is not restricted to linear dynamics” (see [65], p. 4). We highlight three features of this definition:
*A complex interconnected system*: A multitude of variables and timescales which are interwoven, interdependent, and interacting.*Contextualised processes*, *specific to each individual*: Individuals follow meaningfully different change trajectories that develop and change with time.*Not restricted to linear dynamics*: Inputs are not necessarily proportional to outputs, and long periods of apparent stability can precede short periods of rapid change.

## 2. Behaviour Change Mechanisms under Complexity: Three Key Features

In the following three sections, we drill further down into these ideas. In the first, we introduce interconnectedness via interaction-dominant dynamics, which flow from point 1 above; second, we present how idiosyncratic, non-stationary change trajectories lead to non-ergodicity, a technical term for point 2; third, we highlight that the flexibility of complex systems leads to ubiquitous non-linear dynamics as alluded to in point 3. Table 1 provides an overview of these ideas, which are elaborated further in the subsequent sections.

### 2.1. Interconnectedness

When processes in complex systems are not independent, they are said to be coupled. Coupling can be unidirectional (for example, physical activity increases muscle mass but not the other way around), or bidirectional, where the elements of a system simultaneously reinforce or suppress each other over time, demonstrating a type of circular causality (e.g., good performance and rewards). Dynamics in living systems tend to be dominated by synergies (“interaction-dominant causation”) instead of their component parts [41,46,73]. Many psychological and behaviour change theories, at least implicitly, assume the presence of reciprocal causation and intertwined processes (e.g., [74], p. 6), but empirical testing of such processes has, to date, been limited.

As mentioned earlier, within the conventional approach to behaviour change intervention evaluation, researchers commonly employ mediation analyses to examine mechanisms. However, the clean *independent variable*
→
*mediator*
→
*dependent variable* type of path analysis can be misleading when change is in fact driven by self-reinforcing, “autocatalytic” interactions that occur over time. In component-dominant causation, effects follow causes in a billiard-ball fashion, and one variable can change without everything else changing. For example, a study developed with the component-dominant mindset could explore how a specific behaviour change technique, say goal setting, affects behaviour. On the other hand, variables of interest to behaviour change researchers are unlikely to change without affecting a large amount of other related variables [51], producing context-dependent effects [75]. This, too, implies that interaction-dominant causation is a more plausible framework for the behaviour change domain, wherein effects emerge (and are conditional upon) the system’s holistic multivariate dynamics, with everything potentially taking place simultaneously in a circularly causal manner. Interaction-dominant dynamics are also characterised by heavy-tailed distributions [49,66,76] such as the log-normal distribution [77], which are common in psychological data [78,79], as well as the presence of long-range temporal correlations and power-law scaling [72,80,81]. Importantly, interplay happens not just between variables, but also between their temporal dynamics: Processes taking place on fast timescales (e.g., lack of physical activity) modulate slow-timescale processes (e.g., development of obesity, lower energy levels), which then feed back, affecting fast-timescale processes [41].

One way of looking at mutually interacting processes with reciprocal causality is to consider the system as a network. Network science is a well-established field with applications ranging from physiology to the organisation of cities [82], and health [83,84]. An illustrative example comes from the study of depression, where the traditional thinking assumes that a latent factor—depression—causes the symptoms. On the other hand, a network science perspective leads to an alternative view, where the network of mutually interacting symptoms constitutes the phenomenon [85,86]. This approach has provided new avenues into understanding and treating depression, such as locating the symptoms which are most relevant to the activation of the network (i.e., the emergence of depression). In addition, this network approach provides insights into how intervening on specific symptoms might affect the system as a whole, given the dynamic interconnected relationships between symptoms.

Although the network theory of mental disorders [85] aligns with and stems from complexity science, the psychological network models usually associated with the approach [87,88] rely on many assumptions stemming from their grounding in multiple regression, including multivariate normality (i.e., linearity) and stationarity [89]. Despite their valuable richness, they were recently shown to not reliably inform about the underlying system dynamics [90]. Still, the conceptual frameworks such models represent—coupled processes interacting in a system, instead of “root causes” [91]—ought to be the primary ontology considered by behaviour change researchers. In the later section on empirical solutions, we present a recurrence-based network modelling approach to consider when investigating these coupled processes [92].

### 2.2. Non-Ergodicity

To be useful to individuals, processes postulated by psychology should work on the individual level [93]. Whether group-level variation is informative of individual-level dynamics depends on a condition known as ergodicity. Ergodicity has the following properties: “Only if the ensemble of time-dependent trajectories in behaviour space obeys two rigorous conditions will an analysis of interindividual variation yield the same results as an analysis of intraindividual variation […] First, the trajectory of each subject in the ensemble has to obey exactly the same dynamical laws (homogeneity of the ensemble). Second, each trajectory should have constant statistical characteristics in time (stationarity, i.e., constant mean level and serial dependencies)” ([94]; see also [68]).

In other words, in a 100 × 100 spreadsheet, where participants are rows and measurement occasions are columns, calculating an average of values within one column (“ensemble average”), should give the same result as calculating the same statistic from one row (“time average”). For example, in an ergodic process, the mean and standard deviation of each person’s daily minutes of physical activity over a 100-day period would be the same as the mean and standard deviation of 100 people’s daily physical activity minutes measured once. Or, observing that 20% of a given population are smokers, would mean that everyone is a smoker for 20% of their lives. In terms of coupled processes, the correlation between physical activity and intention would be the same in the population measured once, as it is for one person over time.

Going back to the two “rigorous conditions”, the condition of homogeneity almost by definition rules out the behaviour change researcher’s interests, as we are interested in how people (can) change, and it is quite clear that people do not all follow the same behaviour change processes. Indeed, it would seem preposterous to suggest that a process like self-regulation would remain constant across an individual’s life span. The mathematical proof for the non-equivalence of inter-individual and intra-individual data structures was published over a decade ago [95], and recent research has attempted to quantify the threats stemming from lack of group-to-individual generalisability [67]. This preliminary work indicates that even if we could work with “generalisable” ideal random samples from well-defined populations, we would still be committing the ecological fallacy (i.e., drawing individual-level inferences from group-level data) if we wanted to apply our knowledge to individuals.

The second condition, generally referred to as stationarity, is that the statistical properties of these processes must not change over time. In the context of physical activity, the factors that influence behaviour are likely to change over time. For example, the effect of discomfort on physical activity is likely to change in a non-linear manner over time as fitness and tolerance of discomfort fluctuate, not only because of randomness, but as core features of the phenomena itself [96]. The tools most often used in research to analyse behaviour change, such as linear regression, do not account for these kinds of temporal dynamics. This is because temporal cognitive change fundamentally violates the assumption of stationarity.

For the processes underlying physical activity to be considered stationary, the average level of discomfort must remain stable across time for all individuals. In addition, the sequential dependence between repeated measures must be stable [97]. In terms of the relationships between variables, the assumption of stationarity requires that the causal structure which leads to a particular outcome is unchanging across time [98]. Examining behaviour change usually involves an attempt to change the causal structure underlying a behaviour. For example, after making coping plans to tackle barriers to physical activity, the causal relationship between perceived barriers and low physical activity ought to be diminished. Generally, this also means that behaviour can be expected to change as learning and development progress. Stationary data is therefore rare in behaviour change research. This lack of stationarity is rarely acknowledged or (statistically) accounted for in empirical studies evaluating behavioural processes. The result is analogous to the ecological fallacy of taking a population-level mean and extrapolating to individual-level attributes; an average over an individual’s time series describes that individual better than the population-level snapshot, but still might not apply to any particular time period. As a simple example, consider 100 days of data in which a dependence relationship is strongly positive for the first 50, and strongly negative the other: In this case, the average association over the whole time series might be zero, and this misses the abrupt change in this dynamic over time.

Figure 2 illustrates non-stationarity in the case of work motivation, a key feature of occupational health psychology. In these data, (taken from one participant in an observational intensive longitudinal study of work motivation; Heino et al., in prep), the observed relationships between variables shift drastically as the study progresses.

Idiographic science, which tries to unveil person-level processes, does not aim to go inductively from data to universal or statistical laws that hold in hypothetical infinitely large populations [99,100]. Instead, it applies general principles, such as universal properties of complex systems, to study how individuals behave in their particular contexts. Answering more than half a century of calls to expand focus beyond outcomes to processes, new technology in data collection and analysis has now made the idiographic approach possible [101]. The basic solution is to not average individuals and then model the behaviour of the averages, but to first model individuals, and then aggregate those models to search for commonalities [65]. Recent work has made use of methods such as ecological momentary assessment [102] to gather intensive longitudinal data on behaviour and determinants from one or more individuals, which can then be represented as time series. In the case of smoking, analyses of such idiographic data have yielded individualised models that can predict behaviour with stunning accuracy [103,104].

If the mechanisms of behaviour change occur within an individual, then we need to also study them within individuals. However, when we study individual time series data, the methods used in the conventional approach for studying group averages (e.g., pre-post measurements with a long time between them) leave us wanting. Figure 3 illustrates the effects of within-individual sampling rate on perceived trends. When the sampling rate does not match the rate of progression of a phenomenon, a deceptively linear picture of the process might emerge (see also [53], p. 3). The same logic applies if we are studying groups but cannot rely on the means being informative due to a lack of power (as demonstrated in [105]).

In sum, to study individual behaviour change, we need to not only collect intensive longitudinal data on the individual-level, but we must also consider the time evolution of the phenomenon and apply statistical analyses which can accurately model non-stationary data. In the health psychology context, Bolger and Zee [37] argue that not only temporal processes need to be considered, but also the heterogeneity therein. Consistent with the idiographic approach outlined above, every individual may exhibit idiosyncratic dynamics. As we will see next, the possibilities are vast when stepping outside the linear worldview.

### 2.3. Non-Linear Dynamics

The linear methods traditionally used in psychology (e.g., multiple linear regression, ANOVA, and other cases of the general linear model) view psychological phenomena as following gradual changes over time. While sometimes useful as approximations, the assumptions of linear models are usually violated in practice [24]. Furthermore, linear models may be invalid when ceiling or floor effects are present [106,107], or under *hysterisis*, when the temporal direction of a relationship matters for its impact (e.g., prevention is important precisely because it takes more effort to exit the state of having a lifestyle disease, than to enter it) [108,109].

While a reliance on linear models simplifies the analytical approaches needed to explore relationships between variables, it does not contribute to our understanding of how the world works, as “most of everyday life is non-linear” [110] and outside the physical sciences, non-linear systems are “the rule, not the exception” [111]. As an intuitive example, consider that falling from 10 m is likely to kill you, but falling from 1 m does not make you 1/10th dead—in fact, it makes you stronger [112,113]. Or that eating twice the size of a normal meal rarely results in twice the pleasure.

Non-linear dynamics can be very useful albeit unintuitive to grasp, as the world discovered during the COVID-19 pandemic: An exponential growth starting from 20 cases on day 1 with a growth rate of 20% can lead to 4030 cases by day 30, and 81,030 cases by day 45. Reducing this growth rate by a mere one percentage-point would result in approximately 29,000 fewer cases by that time. Theories and methods to understand non-linear change phenomena in individuals can provide different types of answers than linear analyses. The most important factors in predicting behaviour change may not be the strength of a variable’s relationship with behaviour (e.g., regression weights), but rather the type of fluctuation that the variable exhibits in response to an intervention [73,114,115], or how fast the dynamics recover after shocks [116]. 

Another key insight is that while we cannot usually predict what the value of the next observation will be, we can “predict” which system states are possible, and evaluate the risks and opportunities for intervention from there. This can be done with, e.g., simulation approaches, which can potentially “replicate the global tendencies of the dynamics” [117], without being precisely correct about any specific instance. Indeed, exact prediction beyond some short-term horizon is convincingly shown to be impossible in complex and not-so-complex systems across sciences, as indicated since Poincare’s discovery of the famous three-body problem [110]. For a discussion on evaluating the aforementioned prediction time horizon in psychological self-ratings, see [72].

Polynomial regression is perhaps the most commonly used model when linearity is questioned. This method allows for identifying curves that may better fit data on the relationships between variables than a straight line [118], and can also be used to represent non-linear changes that occur over time. Polynomial regression models do not, however, adequately capture the essence of complex systems; non-linear, irregular changes, periodic peaks and plateaus, and with recoveries after negative shocks and deterioration after positive ones [119].

When we consider the situation where all components of a system interact, many features evident in everyday life but ambiguous in linear modelling become salient. Long periods with no discernible changes in outcomes might be followed by short bursts with large shifts. For example, a person’s conscious intention to smoke may remain stable, while social norms keep changing, until one day when a seemingly innocuous event causes the person to quit. When a system finally reaches a “tipping point” (e.g., an individual’s behaviour changes), conventional analytic methods have difficulty determining whether the effect was caused by a critically important incident, or by less obvious, small, cumulative effects over time which preceded the so-called *phase transition*. Obviously, in such situations, the consequences of an incident (i.e., the camel’s back breaking) do not relate linearly to the intensity of the event (i.e., loading the last straw on the camel). This is a common dynamic in complex systems [63], but it is extremely difficult to evaluate if information regarding the system is only available for a few points in time. Intensive longitudinal data are therefore needed.

## 3. Empirical Solutions

To model intensive longitudinal data, models developed within the literature on time series analysis are necessary [65,120]. A time series in this case is a sequence of values representing one variable in one individual, and time series analysis consists of methods for studying time evolution of one or more data generating processes.

The most common modelling framework, lag-1 autoregression, uses one previous time point as input to predicting the next one. In behavioural science, vector autoregression—vectors being sequences of numbers, representing values of variables—is often used to test the effects of several variables on the outcome of interest. One drawback of such autoregressive models is that they assume that there exists an average value around which the process fluctuates, which also motivates the common practice of “detrending”. In detrending, the researcher transforms the data by fitting a linear regression line and continuing the analysis with the residuals, often not taking into account that there can be several trends in subsections of the data (i.e., the trend is non-stationary), which all contribute to what the linear model interprets as normally distributed “errors.” Moreover, the supposed mean value—as well as variance around it—may not remain the same across time (i.e., the level is non-stationary), and the impact of previous time points on future ones is assumed to remain constant [121]. One way to overcome this shortcoming, is to let the parameters in autoregressive models vary across time, leading to the time-varying autoregressive model depicted in Figure 2. But even time-varying autoregressive models operate under the linear regression framework, with its accompanying assumptions, such as normally distributed errors. Furthermore, in Figure 2, we have limited ourselves to investigating the lag-1 relationships, whereas long-range dependencies are common in ecological momentary assessment data [72,92,115,122].

Time-varying autoregressive models are regression-based, and are only appropriate when the dynamics of all variables in the model conform to the required assumptions. Empirical researchers have a wide variety of assumption tests at their disposal. The supplementary website (section available online: https://git.io/JfLmm (accessed on 1 May 2021)) presents a plethora of these tests applied to a sample of 20 individuals collecting data for 9 motivation variables. We can see that many or most time series indeed exhibit non-stationary trends and levels, as well as non-linearities. Additionally, longer time series reject more of the assumptions, as the deviations from assumptions are not necessarily present in small samples, and larger samples confer higher statistical power. This does not suggest that we ought to only gather short time series, as doing so would limit our abilities to detect deviations from assumptions and generalise to data outside the sample.

There are many ways to study non-linear change processes in complex systems. Behavioural researchers may find the generalised logistic model [107] a good starting point. This method produces readily-interpretable parameters indicating the floors and ceilings of the variables intervened upon, as well as the growth rate and timing of changes. Researchers may also be interested in identifying critical transformations taking place in a system (e.g., a person’s motivational system). In complex systems, these state transitions may be preceded by warning signs such as increased turbulence (quantified as e.g., dynamic complexity; [123]), or critical slowing down (i.e., heightened autocorrelations in a time series), before (re)lapses occur [124,125]. In clinical psychology, intensive monitoring of psychopathological symptoms has allowed researchers to examine symptoms’ variability, autocorrelations and other indicators of dynamics during interventions. This has yielded considerable advances in the prediction of phase transitions between adaptive and maladaptive psychological states [58,126,127,128]. A conceptual replication in a population undergoing a weight loss intervention [129] recently found that sudden drops in physical activity levels could be predicted by the emergence of erratic fluctuations in day-to-day physical activity. While the presence of critical fluctuations is a key indicator of the effectiveness of psychotherapy for mood disorders [58], similar investigations have not yet been undertaken in other behaviour change contexts.

Multilevel models are often proposed as a method to handle nested timescales and temporal dynamics. It should be noted though, that such models assume, e.g., that individuals depart from group-level means according to some known distribution—and if distributional assumptions are incorrect, so are the results [100]. For a use case of complexity-informed multilevel modelling, see [129]. 

In the next section, we present one family of analysis methods, recurrence quantification, that is suitable for analysing longitudinal data sets while making fewer a priori assumptions.

### Modelling Complex Time Series Data with Recurrence-Based Analyses

Recurrence quantification analysis, unlike regression-based methods, makes no assumptions about distributional shapes of observations or their errors, about linearity, nor about the time-lags involved. Researchers can therefore use it to explore the dynamics of a phenomenon, obtaining robust visually-intuitive information about the organisation of a system. Recall from Table 1 that, in complex systems, the organisation of components can be more important than the components themselves. 

Recurrence networks display relationships between multivariate observations in a time series in an intuitive way. The results of a multidimensional recurrence quantification analysis can be thought of as displaying a type of multivariate “correlation”, indicating the occasions in a time series that repeat a previously observed pattern. This could be patterns of single values or combinations of values of different variables, analogous to a system state. These patterns or configurations can be thought of as the attractor states towards which the system is drawn.

An in-depth walkthrough of the analysis with code is provided in the supplementary website (see section available online: https://git.io/JfLs3 (accessed on 1 May 2021)), and a tutorial is available in [130], hence we will be brief in the background and focus on the results. Data for the demonstration below comes from a single participant, who at each time point, completed six questions about their self-determined motivation. A more detailed exposition of the data is found at the supplementary website (see section available online: https://git.io/JfLmQ (accessed on 1 May 2021)).

Figure 4 demonstrates a multidimensional recurrence network, where each point is a measurement occasion. Lines between measurement occasions indicate recurrences of a system state, in this case, a “motivation profile” consisting of the six motivation-related questions. We can see that most of the recurrent states take place in the second half of the data. In addition, patterns that appear only once (white dots) take place exclusively in the first half of data collection. Had we only measured the first 50% of observations, many of the recurrent system states would have been missed. This network demonstrates that this person’s motivational system gravitates between several observable recurrent states.

While Figure 4 shows how various states recur over time within the system, Figure 5 depicts the actual values of variables within each recurrent state. With this information, one might attempt to assign qualitative meaning to the observed recurrent profiles. In this data, we can observe that about a fifth of the participant’s responses fall into a relatively balanced profile (category 1st, yellow), while ca. 15% display what self-determination theory [132] would consider an “optimal” motivation profile—high in autonomous forms of motivation and low in controlled ones (categories 3rd and 4th, purple and dark blue, respectively).

We can collapse the information in Figure 4 to the percentages with which one state follows another, demonstrating a *transition network*. It answers the question “If you are in state x, what is the probability of transitioning to state y?” Figure 6 (panel A) depicts the relative frequencies with which each state precedes the others. If the system displays linear dynamics, then the matrix should be roughly symmetrical, with similar values observed above and below the diagonal drawn from the bottom left cell to the top right cell. Panel B of Figure 6 presents the same information as a network.

To distinguish whether the results reflect a non-linear data structure, or whether they are merely a product of randomness, researchers can use a technique called surrogate data analysis [133]. In this method, temporally disordered versions of the data—called “surrogates”—are created, and the observed data is then compared to those. The surrogates represent the hypothesis that the data were generated by a rescaled Gaussian linear process. By analysing the surrogates, we ask whether the data can be understood to have arisen from a process that is essentially stochastic and linear instead of highly interdependent and non-linear. The analysis indicates that it would indeed be very unlikely to see these results, if the dynamics were Gaussian. Surrogate analysis of this dataset is presented in the supplementary website (section available online: https://git.io/JqRTQ (accessed on 1 May 2021)).

Following the idiographic approach outlined earlier, this information could now be used to develop a personalised intervention. The designer of such an intervention—possibly the person themself, using self-enactable behaviour change techniques [134]—has several new perspectives to consider: How could transitions to the “optimal” profiles (3rd and 4th) be increased? Could these optimal states be made more “sticky”, and the 1st state less so (as indicated by self-loops in Figure 6, panel B)? Could just-in-time adaptive interventions [56]—such as prompts on a mobile device—be used to inform the person of the state they are in, and prompt enactment of techniques that can help them stay in or leave that state?

To summarise: Having looked at all the time scales instead of just the previous time points, while not restricting ourselves to linear dynamics, we observe the features outlined in Table 1. The recurrent states of the system are connected across time, demonstrating interconnectedness instead of independence. They are not equally spaced in time, demonstrating non-stationarity and hence non-ergodicity. In addition, going from state a to state b does not generally happen with the same probability as the reverse, hence demonstrating non-linear dynamics. These features offer potent information for formulating interventions and understanding the dynamics at play in a system, but would be overlooked by more traditional analysis methods.

## 4. Discussion

Applied behavioural sciences have always studied phenomena, like behaviour change mechanisms, which take place within complex ecological systems [135]. Past efforts to understand these phenomena used linear models, even though the tools of complexity science would have been more appropriate [136]. Behavioural scientists have an opportune moment to start considering complexity, as the field of behavioural intervention research is now taking committed first steps in this direction [64,75]. There is a growing interest toward intervention programme theories that explicitly model complexity, such as recursive causality, disproportionate relationships, “tipping points”, and emergent outcomes [137]. In addition, novel analytical methods that are compatible with complexity science are continually being developed [131].

By examining intensive longitudinal data, we have shown similar results to a plethora of studies, which find complex dynamics in ecological momentary assessment data [72,115,122,138]: Non-linear, non-ergodic, non-independent dynamics, which defy traditional assumptions. Moreover, the empirical case example of a single person shows the importance of observing change over long periods of time: There are several recurring “ways of being”, patterns of experience which change dynamically within the individual. These recurrence patterns, and the nature of states that recur, are likely to differ across individuals. By using multivariate recurrence-based methods [92], researchers are free to examine conceptually overlapping variables from multiple theories in the same analysis [139], as such methods do not require partialing out variance. This makes intervention process evaluation possible from a more holistic perspective—that is, looking for changes in, e.g., attractor states or complexity measures.

Critically appraising the often hidden assumptions of models, especially in the context of complex systems such as human behaviour change interventions, is necessary for understanding the phenomena of interest and building a credible science. Researchers who study stable phenomena and who only wish to draw group-level inferences (e.g., to select promising public health interventions) are probably best served with traditional models. This is rarely the case for psychologists and behaviour change intervention researchers who wish to understand time- and context-embedded change mechanisms. For theory to advance, assumptions need to be justified: We cannot conclude, both, that our models for empirical testing omit crucial facets of reality, and at the same time imply real-life consequences. We propose that a more fruitful approach would be to investigate coupled processes with individual-level psychological data from intensive longitudinal designs, and to use analyses that are reasonably free from assumptions regarding independence, ergodicity and linearity. 

By studying what other sciences know about change processes in complex systems and replicating studies where the ideas have been applied to human behaviour change, researchers can work towards uncovering more general principles of behaviour change. As Molenaar [140] pointed out, “the set of person-specific time series models thus obtained then can in the next step be subjected to standard analysis of inter-individual variation in order to detect subsets of subjects who are homogeneous with respect to particular aspects of the dynamical laws concerned”. In other words, information obtained from individual-level studies of dynamic patterns can then possibly inform models of larger groups, leading to better (or at least humbler and more nuanced) social scientific theories [141]. Generating theory in this way would answer calls to address the issue of time more clearly in theories of health behaviour [36,37]. It could also lay the foundation for more formal theories of behaviour change to be developed [50,90,142], as these typically hypothesise how relationships between variables unfold over time, and a more coherent correspondence between theoretical cycles and empirical cycles in behaviour change research [143].

## 5. Limitations

The field of complexity science and aligned novel methods is fast-moving, with new developments always on the horizon. However, there remain many practical and methodological barriers to fully embracing the complexity perspective in behaviour change research. Many of these barriers relate to data collection. While the development of smartphones and wearable devices for ambulatory assessment allow the convenient collection of intensive longitudinal data, there are few stable and user-friendly open-source options. This has resulted in large variability in the data collection tools used to produce intensive longitudinal data [144]. Ensuring good adherence to these forms of data collection can be a challenge for researchers. For participants, adapting to intensive assessment is a behaviour change in itself—particularly if they are required to use a specific device or smartphone application. Although measurement burst designs [145] might alleviate some challenges, they bring about other, perhaps graver ones, such as a mismatch between the sampling rate and the time scales in which the phenomena of interest unfold.

Long time series data can be time-consuming and effortful to collect. It also creates a much greater burden on participants than traditional questionnaires and fewer time points. However, in behaviour change research and health psychology, much of the core research interests of our theories—influences on behaviours—have traditionally been subjective factors (e.g., sense of self-efficacy, motivations and motives, outcome expectancies), only—by definition—accessible via self-report. This presents an undeniable practical challenge, along with the fact that observations often need to be spread equidistantly in time, allowing one individual to collect only 1–2 data points per day. Still, examples of more than a hundred time points being collected are found from weight loss maintenance [146] to psychotherapy [127], with some studies collecting more than 1000 observations per participant [115].

Several methodological challenges for the study of dynamic systems in behavioural science have been identified [147], including measurement reactivity, the optimal choice of measurement intervals, and measurement quality. To properly address measurement reactivity, it is necessary to know whether the anticipation of measurement or the self-monitoring process itself (or both) interact with the outcomes of interest. Choosing an optimal measurement interval requires knowledge of the timescale of the behaviour change dynamics, which is rare. As regards to measurement quality, we still lack a comprehensive approach to developing and establishing the quality of momentary measures of psychological constructs. Ensuring the validity and reliability of these measures can be difficult due to the requirement to use few items, not to mention that the questionnaire scales are themselves bounded, whereas experience hardly is. One solution for this is to inspect change profiles of responses [92] instead of raw scores. Another solution would naturally be tapping into wearable data; for example, electronically-activated recorders [148] are maturing as a technology, and complexity methods have already been applied to physical activity data during a weight loss intervention [129]. 

## 6. Conclusions

When a study finds that variables have explained an unsatisfactory proportion of behaviour, researchers often follow the pattern seen in social and organisational sciences and conclude that either: “(a) significant, explanatory variables have been omitted from the study, (b) the measurement instrument is too imprecise and ‘rough’, or that (c) the random or stochastic part of the problem has overwhelmed the patterned part” [149]. However, if the result stems from a statistical model that makes unfounded assumptions regarding independence, ergodicity and linearity, is it any wonder that it fails to satisfactorily describe reality? In this paper, we have attempted to show that many common modelling strategies fail to adequately capture real-world dynamics of behaviour change, and that a change in approach can advance our understanding of behaviour and behaviour change processes. Behaviour change researchers should further utilize intensive longitudinal designs to collect individual-level psychological and behavioural data from participants, and should utilise analytical methods that are reasonably free from assumptions of independence, ergodicity and linearity. This has practical implications from replicability to outcome and intervention selection. In our view, further embracing complexity science and its methods will advance research on behaviour change and could unearth new evidence of the dynamics of behavioural processes.

## Figures and Tables

**Figure 1 behavsci-11-00077-f001:**
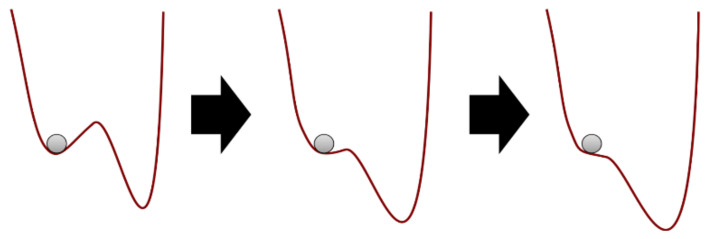
Evolution in attractor landscape: An intervention moulds a system, making it less stable, hence easier for the ball to move from current state (**left**) to another one (**right**). Alternatively, an intervention—or random events—can jolt the system over the ridge, i.e., a tipping point.

**Figure 2 behavsci-11-00077-f002:**
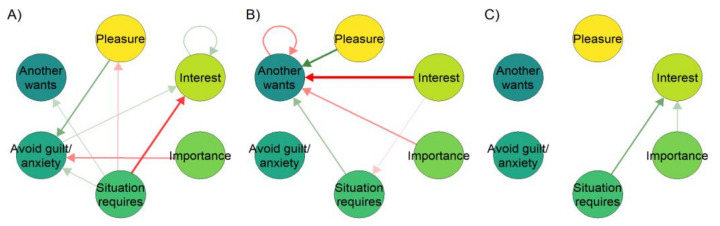
Relationships between a single participant’s motivational variables varying in time (time-varying autoregressive model). Networks represent relationships between variables around the time points where 10% (panel (**A**)), 50% (**B**) and 90% (**C**) of the study had been completed. An arrow from one variable to the next means the former predicts the latter at the next time point; green for positive and red for negative correlation. If a stationary model was used, all periods would be collapsed to a single result, creating the impression that the relationships were homogeneous across the study period. Although this temporal variability can be due to, e.g., changes in how the participant answers the questions (boredom, shifting perception of the items, etc.), or poor reliability of the measures, complexity theory would also guide us to expect that in very concrete reality, the direction and strength of relationships can shift over time and differ based on the state a person resides in. As an example, the relationships between motivational variables during behaviour change initiation phase, may differ from the relationships during the maintenance phase.

**Figure 3 behavsci-11-00077-f003:**
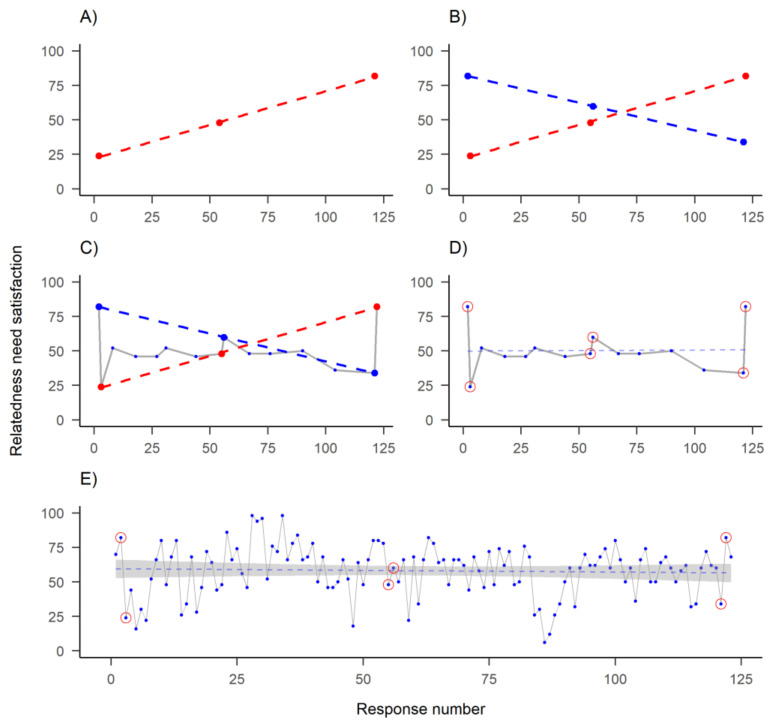
One of the time series recorded by the participant featured in previous figure. Dots indicate answers to a visual analog scale question on their relatedness need satisfaction, as posited by self-determination theory (*y*-axis), measured on different time points (*x*-axis): (**A**) Measuring three time points—representing conventional evaluation of baseline, post-intervention and a longer-term follow-up—shows a decreasing trend; (**B**) Same measurement on slightly different days shows an opposite trend; (**C**) Measuring 15 time points instead of 3 would have accommodated both observed “trends”; (**D**) New linear regression line (dashed) indicates stationarity and (**E**) Including all 122 time points, a more complete picture of the dynamics emerges.

**Figure 4 behavsci-11-00077-f004:**
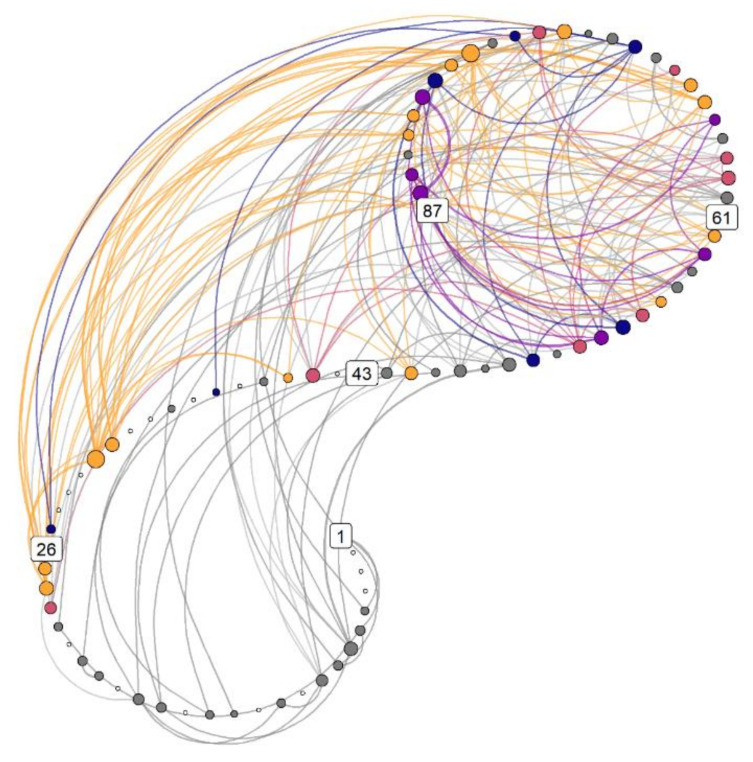
Weighted multidimensional recurrence network. Each circle (“node”) is a measurement occasion, numbers indicate their running number, and colors represent different motivation profiles. These profiles are configurations of six variables, and can be conceived of as attractors. Lines indicate the same motivational state reoccurring at a later time point. Yellow nodes indicate configurations connecting to that with the highest strength centrality (i.e., number of connections weighted by the similarity of the connected nodes), red nodes connect to the second strongest configuration which is not connected to the strongest, followed by purple and blue. Grey nodes depict uncategorised configurations which occur at least twice, and white ones depict the configurations, which only occur once. Nodes that are larger have higher strength centrality. Drawn with R package casnet [131].

**Figure 5 behavsci-11-00077-f005:**
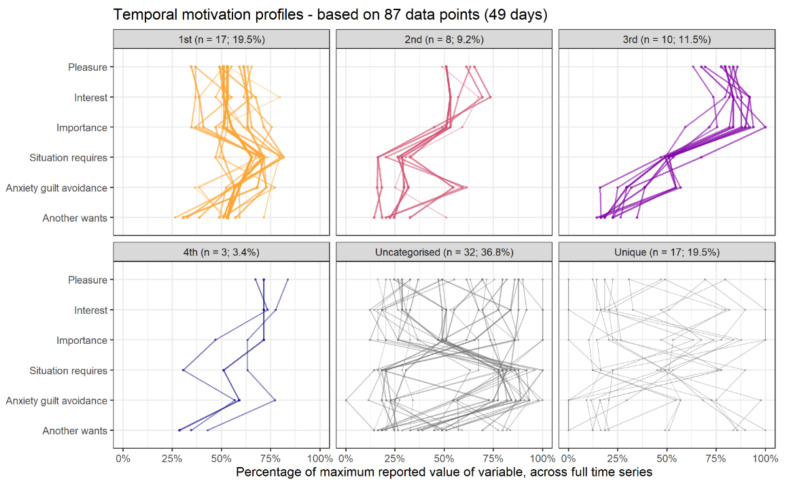
Main profiles corresponding to the colors indicated in the previous plot. See supplementary website (section available online: https://git.io/JfLmS (accessed on 1 May 2021)) for a thorough exposition.

**Figure 6 behavsci-11-00077-f006:**
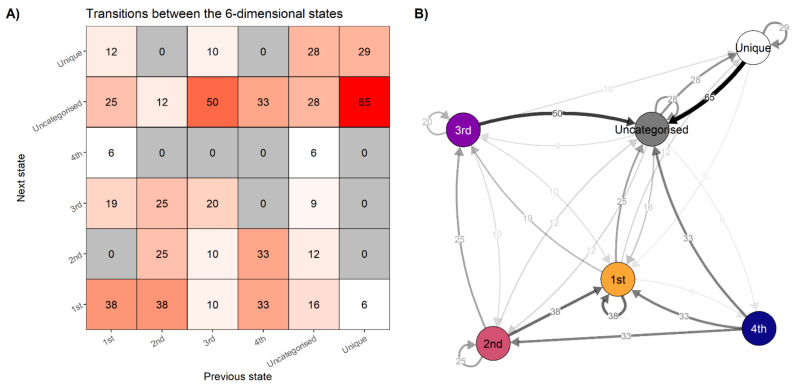
Transitions between states. Panel (**A**). Percentages with which each state precedes the others. If the system is in the configuration labelled 1st, based on the relative frequencies of observed transitions, there is a 38% chance it stays in the same configuration, and a 25% chance it transitions to one of the uncategorised states—that is, states that are less strong than the state labelled 4th, but which appear more than once. Note that the zeroes do not signify this transition is impossible, only that it did not appear once during the data collection period. Columns may not sum to 100 due to rounding. Panel (**B**). Data from panel (**A**) represented as a transition network.

**Table 1 behavsci-11-00077-t001:** Three common features of complex systems, with recommendations for behaviour change research.

	Interconnectedness	Non-Ergodicity	Non-Linear Dynamics
Description	The structure of a system—how it is organised and the relationships between its component parts—can matter more than the component parts themselves. This includes interconnectedness of different variables such as attitudes or perceived norms, as well as that of their temporal dependence; dynamic dependencies of complex systems are not restricted to one or a few previous time points [41,46,66].	Psychological processes are non-stationary and heterogeneous, hence non-ergodic (group-level measurements do not correspond to those of individuals in time). This means within-individual processes cannot, in general, be inferred from between-individual data. The lack of group-to-individual generalisability implies a threat to validity of results in many areas of science [67,68,69].	In a linear progression of a phenomenon, the whole is exactly the sum of its parts: You can calculate how much each influencer of behaviour changes, and add them together to get the total effect. Non-linearity occurs when a system’s inputs are disproportionate to its outputs. For example, an effect might be imperceptible for a long time, then explode (as in exponential growth), or suddenly switch states upon reaching a threshold [70,71,72].
Main lesson	Dynamic, intertwined processes do not exist in a vacuum. They are always co-dependent and cannot be partialed out into variance components without losing essential information on how the system as a whole operates.	Drawing individual-level inferences from group-level data (the ecological fallacy) leads to misleading or incorrect inferences regarding individual behaviour. A statistical relationship in the population may not hold for any of the individuals.	Viewing the world solely from the lens of linear phenomena and relationships, leads to missed opportunities and misunderstood impacts of interventions.
Recommendations for the research community	Move from traditional regression-based approaches, which are inspired by component-dominant, additive dynamics, to methods developed for interaction-dominant dynamics, able to cope with multiplicative effects and heavy-tailed distributions.	Move from large-sample research with many variables and many people but few time points (one model per sample), to N-of-1 and intensive longitudinal time series designs, with usually fewer people and variables, but more data per variable (one model per individual).	Move from linear approximations with the illusion of predictability, to methods that can accommodate non-linear patterns and disproportionate influences.

## Data Availability

Data is available on the supplementary website (section available online: https://git.io/JfLmQ (accessed on 1 May 2021)).

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
