# Peer review of "Studying Behaviour Change Mechanisms under Complexity"

_behavsci, 2021, doi:10.3390/bs11050077_

Round 1
Reviewer 1 Report
First of all, thank you very much for the opportunity to review the manuscript titled "Studying behaviour change mechanisms under complexity".
In general, it is a work of great quality and practical relevance. At a formal level it is perfectly structured and in terms of content, I consider that the wording should be lightened since, at times, it is confusing and it is difficult to follow the argument of the work. In addition, in my opinion, I consider that it should show greater emphasis on the practical implications of the study, that is, underline the relevance and novelty of the study compared to what has been previously published on the subject.
Best wishes for Authors.
Author Response
(R1) At a formal level it is perfectly structured and in terms of content, I consider that the wording should be lightened since, at times, it is confusing and it is difficult to follow the argument of the work.
- Thank you for this comment. We have gone through the text and used the Hemmingway app to simplify the wording as much as possible. In particular, we broke down some longer sentences into their component parts, and tried to more often use the active voice. We hope that this will make the text clearer and easier to digest for readers.
(R1) In addition, in my opinion, I consider that it should show greater emphasis on the practical implications of the study, that is, underline the relevance and novelty of the study compared to what has been previously published on the subject.
- This is a really good point. We agree that the paper needed to better emphasize how the methods it describes can be used in practice. To improve this aspect of the paper, Section 3.1 has been edited to include this information, including suggestions for the types of questions researchers might be able to investigate with each method. We think that this will make the practical implication more salient for readers.
Reviewer 2 Report
I was a previous reviewer on this manuscript, and saw considerable strengths to this paper. I did raise a handful of questions, which the authors adequately addressed in this revision. It is an interesting and important paper and I think it will be quite useful for readers of this journal.
Author Response
We thank the reviewer for the efforts put into evaluating this manuscript. As it also appeared in the other reviewers' comments, we have underwent an English language / spell checking process.
Reviewer 3 Report
The authors addressed all my comments, although they only integrated very few of my recommendations. If the manuscript is sent out for another round of revisions, I recommend that the authors integrate the key arguments presented in the response to reviewers in the manuscript as I'm likely not the only researcher asking these questions.
Author Response
(R3) The authors addressed all my comments, although they only integrated very few of my recommendations. If the manuscript is sent out for another round of revisions, I recommend that the authors integrate the key arguments presented in the response to reviewers in the manuscript as I'm likely not the only researcher asking these questions.
- Thank you for this important comment. We agree that it is important to make this dialogue a part of the final article, and we have implemented changes to make this more transparent for readers. Specifically, we have separated the issues regarding simulation and predictability, as well as multilevel models, as their own paragraphs with additional explanation based on our response to your previous comments.
- These can be found on the following pages:
- p. 10: “Another key insight is that, while we cannot usually predict what the value of the next observation will be, we can “predict” which system states are possible, and evaluate the risks and opportunities for intervention from there. This can be done with e.g. simulation approaches, which can potentially “replicate the global tendencies of the dynamics” [150], without being precisely correct about any specific instance. Indeed, exact prediction beyond some short-term horizon is convincingly shown to be impossible in complex and not-so-complex systems across sciences, as indicated since Poincare’s discovery of the famous three-body problem [110]. For a discussion on evaluating the aforementioned prediction time horizon in psychological self-ratings, see [72].”
- p. 12: “Multilevel models are often proposed as a method to handle nested timescales and temporal dynamics. It should be noted though, that such models assume e.g. that individuals depart from group-level means according to some known distribution – and if distributional assumptions are incorrect, so are the results [100]. For a use case of complexity-informed multilevel modeling, see [128].”
Changes made in addition to the aforementioned and the English language editing:
- Added missing funding source for MH: “MH was also supported by Gyllenberg Foundation (grant number 5177).”
- Bullets on p. 5 outlining the three features of the complexity approach to behaviour change, now switched to numbers (as originally intended and referred to in text)
- “As a simple example, think 100 days of data in which a linear dependence relationship is strongly positive for the first 50, and strongly negative the other; you might observe the average correlation over the whole time series to be zero.” → The word correlation on p. 8 is changed to “association” and the word linear is removed, as correlation has features prohibiting inference such as this and the abruptness of the change would be peculiar under linear dynamics.
- We removed the mention of non-linear scaling and space-filling as especially the latter was not discussed in the paper but only referenced in passing. This was replaced with the sentence “Despite their valuable richness, they were recently shown to not reliably inform about the underlying system dynamics [90]” on p. 6. Accordingly, we also replaced reference #90 (which used to be the one for space filling) with a more fitting, novel one.
- Caption in Figure 2: “time-varying autoregressive model” → “time-varying vector autoregressive model”
- First instance of “Idiographic science” italicised for the importance of the concept
- “An exponential growth starting from 10 cases” → “An exponential growth starting from 20 cases”
- dots in front of headings
- “variatiomeasun” → “variation”
- Inserted quote marks around the second predict, as it could be argued this is not really prediction, given that not all the possible states will ever be materialised: “Another key insight is that, while we cannot usually predict what the value of the next observation will be, we can “predict” which system states are possible, and evaluate the risks and opportunities for intervention from there.”
- Some of the text was left without updating, after the bug outlined in previous revision was fixed in the analysis code.
- Percentages in Figure 6 caption updated to match the figure
- “the “optimal” profiles (2nd and 4th)” → “the “optimal” profiles (3rd and 4th)”
This manuscript is a resubmission of an earlier submission. The following is a list of the peer review reports and author responses from that submission.
Round 1
Reviewer 1 Report
Thank you for the opportunity to review the manuscript entitled "Studying behaviour change mechanisms under complexity”.
This article illustrates limitations of these standard tools, and considers the benefits of adopting a complex adaptive systems approach to behaviour change research. The manuscript submitted for review examines a topic of great relevance in the field of behavioral sciences. The topic is very important and the conceptual analysis made in the text is quite deep. The literature consulted is quite current (which is a strength for your work.). I would like to thank the efforts by the authors of the manuscript and congratulate them on the work. Overall, the writing is clear, the goals are well described, the introduction should explain the objectives of the study based on the review of the previous literature and the conclusions are properly made and presented. I consider that the constructs proposed in the abstract of the work are quite well explained. Therefore, the manuscript brings significant knowledge of the scientific literature so and still covers existing gaps in the field. On a formal level, the manuscript should be better structured and it would be convenient to reorganize the information by adding new headings, the references comply with the rules and the DOI is added. The work is ambitious and the results confirm the most of the hypotheses and the relevance and potential of the work is therefore recognized. Personally, I would emphasize the novelty of the work compared to previous studies and would add more importance in the practical implications. Finally, I wish the authors the best in continuing this line of research.
Best wishes for Authors.
Author Response
Reviewer 1:
Thank you for the opportunity to review the manuscript entitled "Studying behaviour change mechanisms under complexity”.
This article illustrates limitations of these standard tools, and considers the benefits of adopting a complex adaptive systems approach to behaviour change research. The manuscript submitted for review examines a topic of great relevance in the field of behavioral sciences. The topic is very important and the conceptual analysis made in the text is quite deep. The literature consulted is quite current (which is a strength for your work.). I would like to thank the efforts by the authors of the manuscript and congratulate them on the work. Overall, the writing is clear, the goals are well described, the introduction should explain the objectives of the study based on the review of the previous literature and the conclusions are properly made and presented. I consider that the constructs proposed in the abstract of the work are quite well explained. Therefore, the manuscript brings significant knowledge of the scientific literature so and still covers existing gaps in the field. The work is ambitious and the results confirm the most of the hypotheses and the relevance and potential of the work is therefore recognized.
- Thank you for your efforts in reviewing this manuscript and for your kind words about its possible contributions to the literature.
On a formal level, the manuscript should be better structured and it would be convenient to reorganize the information by adding new headings, the references comply with the rules and the DOI is added.
- We have now added DOI for items for which it could be retrieved.
Personally, I would emphasize the novelty of the work compared to previous studies and would add more importance in the practical implications.
- We have added a mention of the practical implications in the discussion section, including this passage in the conclusions: “Behaviour change researchers should further utilize intensive longitudinal designs to collect individual-level psychological and behavioural data from participants, and should increasingly analyse such data with methods that are reasonably free from assumptions of independence, ergodicity and linearity. This has practical implications from replicability to outcome and intervention selection. In our view, further embracing complexity science and its methods will advance research on behaviour change and could unearth new evidence of the dynamics of behavioural processes.”
Finally, I wish the authors the best in continuing this line of research.
Best wishes for Authors.
Other changes:
- In the supplementary website, we had a figure depicting different centrality indices calculated from the transition network. As we do not introduce centrality indices, and their application to transition networks likely needs some additional work in order to be a more mathematically robust procedure, we have decided to omit the figure.
- In the supplementary website’s recurrence quantification section, we now also present robustness analysis for changing from Euclidean distance to Chebyshev distance.
- We discovered a small inconsistency in the approach to classifying the attractor states: The “1st” attractor was defined by node degree instead of strength centrality. While node degree and strength centrality correlate > 0.95 (see new section on surrogate data analysis on supplementary website), defining the 1st attractor by strength centrality instead of degree changes the results slightly
Reviewer 2 Report
In “Studying behavior change mechanisms under complexity” authors Heino et al. provide a compelling case for adopting a complex systems approach in conducting behavior change research, and they introduce readers to empirical solutions for doing so. This impeccably written manuscript provides a thorough, state-of-the-art review of complex adaptive systems, particularly in the context of behavior change, and it does an excellent job explaining difficult concepts. Aptly chosen examples, both in the text and in the figures are very helpful to the reader. I think this paper will be of interest to those who know nonlinear dynamical systems analyses well…but in domains outside of behavior change (e.g., basic human movement science), and it will be particularly exciting (for both theoretical and methodological reasons) for any behavioral change researchers who are accustomed to doing conventional general linear model/ANOVA/regression type analyses. The article alone will be a wonderful resource for researchers (not to mention the authors’ supplementary data and codes—a goldmine for folks hoping to do such analyses!).
In sum: this article will make a significant contribution to this journal. I have very few suggested changes regarding the manuscript. In the comments below I first elaborate on my positive comments above, then a few minor comments or typos follow those.
Particular strengths:
- The authors do a very good job explaining the limitations of the conventional approach in the introduction, and in section 1.2, and in Figure 3.
- It is so difficult to explain complex systems to researchers who are only familiar with conventional approaches—and it is particularly challenging to do that while simultaneously applying the concepts in the specific domain of those researchers. (I would love to hand this paper to an advanced grad student in behavioral change who knows nothing about complexity and find out how well they grasp the concepts by just reading this…and how much it might spark novel research ideas.) Usually even when I read text that does a good job explaining concepts such as “interaction-dominant dynamics”, synergies, non-ergodicity, and so forth, those concepts are explained using examples fairly distant from the researchers’ domain, and only then in a separate section do the writers tackle some examples of how to apply it in a given domain. What the writers have done here is so much harder than that, and has much more impact. Throughout the paper I have scribbled notes re. good examples used, how well they are applied to the domain, or just places where I liked the language used. Here are numerous examples: lines 95-99: distinguishing between complexity and complicated; lines 117-118; interesting point, lines 150-151; good point lines 164-165; lines 165-166; lines 174-183: good summary; lines 224- 229 (depression domain); lines 243-252: well-explained; example: lines 287-292; Figure 3: this illustrates authors’ points well; lines 339-340; page 10 (through end of section 2.0): good examples of hysteresis, exponential growth, explanation of phase transition (and again in line 420-429—fascinating application of critical slowing/increased variability that I did not know), and in line 366-371: good job explaining to researchers who’ll be familiar with using polynomial variables to describe a curved line relationship, why that cannot not adequately capture many features of nonlinear systems; Figures 4 to 6: very useful; lines 488-495; lines 532-534; Table 1: excellent; good summary in Discussion. The empirical section was great as well.
Thoughts/Questions/Comments:
- In general, the authors did a very good job using data and explaining just the minimal required about that data such that the reader can follow the figures displaying the profiles, the recurrence network, and the transition network. For Figure 2, although the figure does illustrate the point (why a stationary model would have missed the shifting structure regarding motivational variables), I think it will be natural for readers to try to make sense of what the shifting structure means over time. (Is this good/bad? Shift in usefulness of the separate items as students get bored answering them for the trillionth time? Some external factor (“history”) occurring that leads ultimately to simplified structure? Do these shifts just signal measures with poor reliability somehow?). That effort is likely to be frustrated because of the figure standing alone without a theoretical context. However, I can also see the drawbacks to flow if the authors have a long digression here in the text! (As a contrasting example, enough hints regarding context + interpretation—lines 470-472 + top of page 14—are offered to allow an interested reader to better understand the implications of Figure 5 and 6.)
- With regard to the limitations section: the authors seem to imply all measures in behavioral change research are momentary self-reported measures of psychological constructs. Given the wearables revolution, I would think that some of these limitations could be overcome if data from physiological, movement (activity), or recorders (E.A.R.) are theoretically informative and thus included as well.
- Over half of the data points in Figure 5 were either uncategorizable (about a third) or unique (nearly a fifth). That suggests quite a bit of data that is difficult to interpret. Given how bottom-up nonlinear dynamical analyses are, one concern I have long heard from researchers only familiar with conventional analyses is whether the analyses would find “something” even if the data were nonsense—that it will capitalize on noise or error, and find some kind of structure to fit the data from whatever one feeds into it. The challenge is how does a researcher validate that a structure that is found is different from what one would find by chance alone; isn’t this so post hoc rather than theory-driven? In some kinds of analyses (e.g., cross-recurrence quantification looking at how much one member of a dyad’s movement coordinates with their partner) there are analyses that can be done to give a sense of what results would result from nonsense data. For instance, in that example, one can do “pseudo-pair” analyses to show that if one mismatches the movement data from one member of a dyad with that of a member of a different dyad, the percent recurrence is substantially lower. I wonder if the authors might address the question of validating that the structures found in analyses mean anything or if SOME kind of structures would emerge even if there was no real “there” there (e.g., if the data were shuffled). At least some areas of psychology are famously skilled at coming up with a post hoc interpretation of whatever patterns appear in the data, after all.
- This question follows closely, to my mind, from the last point. I think one challenge for good nonlinear analyses (and interpretation) is that it seems to me that stronger theorizing might be required for successful, sophisticated nonlinear analyses; in contrast I feel one might be able to get by with less theoretical sophistication if one is relying on familiar regression analyses for instance. One worry I have about this excellent paper follows from how easily the authors convey a deep theoretical grasp not just of nonlinear complex systems but also theoretical bases of behavioral change interventions. But if someone uses nonlinear techniques, with little theoretical depth, the problems I mention in the previous comment might be particularly concerning. For me it comes through just how well thought out the theoretical side is for these authors (e.g., lines 488-495), but I don’t know if that will be apparent to a casual reader. Because the authors convey the theorizing side so smoothly that part of the work might look easy to a novice who would not know how much more imperative it is to do good theorizing in advance.
Typos:
Sentence in line 359-360: remove “to” after “provide”.
Line 326: in “collected by the participant” change “by”.
Line 536: remove extra “are”.
Author Response
In “Studying behavior change mechanisms under complexity” authors Heino et al. provide a compelling case for adopting a complex systems approach in conducting behavior change research, and they introduce readers to empirical solutions for doing so. This impeccably written manuscript provides a thorough, state-of-the-art review of complex adaptive systems, particularly in the context of behavior change, and it does an excellent job explaining difficult concepts. Aptly chosen examples, both in the text and in the figures are very helpful to the reader. I think this paper will be of interest to those who know nonlinear dynamical systems analyses well…but in domains outside of behavior change (e.g., basic human movement science), and it will be particularly exciting (for both theoretical and methodological reasons) for any behavioral change researchers who are accustomed to doing conventional general linear model/ANOVA/regression type analyses. The article alone will be a wonderful resource for researchers (not to mention the authors’ supplementary data and codes—a goldmine for folks hoping to do such analyses!).
In sum: this article will make a significant contribution to this journal. I have very few suggested changes regarding the manuscript. In the comments below I first elaborate on my positive comments above, then a few minor comments or typos follow those.
Particular strengths:
The authors do a very good job explaining the limitations of the conventional approach in the introduction, and in section 1.2, and in Figure 3.
It is so difficult to explain complex systems to researchers who are only familiar with conventional approaches—and it is particularly challenging to do that while simultaneously applying the concepts in the specific domain of those researchers. (I would love to hand this paper to an advanced grad student in behavioral change who knows nothing about complexity and find out how well they grasp the concepts by just reading this…and how much it might spark novel research ideas.) Usually even when I read text that does a good job explaining concepts such as “interaction-dominant dynamics”, synergies, non-ergodicity, and so forth, those concepts are explained using examples fairly distant from the researchers’ domain, and only then in a separate section do the writers tackle some examples of how to apply it in a given domain. What the writers have done here is so much harder than that, and has much more impact. Throughout the paper I have scribbled notes re. good examples used, how well they are applied to the domain, or just places where I liked the language used. Here are numerous examples: lines 95-99: distinguishing between complexity and complicated; lines 117-118; interesting point, lines 150-151; good point lines 164-165; lines 165-166; lines 174-183: good summary; lines 224- 229 (depression domain); lines 243-252: well-explained; example: lines 287-292; Figure 3: this illustrates authors’ points well; lines 339-340; page 10 (through end of section 2.0): good examples of hysteresis, exponential growth, explanation of phase transition (and again in line 420-429—fascinating application of critical slowing/increased variability that I did not know), and in line 366-371: good job explaining to researchers who’ll be familiar with using polynomial variables to describe a curved line relationship, why that cannot not adequately capture many features of nonlinear systems; Figures 4 to 6: very useful; lines 488-495; lines 532-534; Table 1: excellent; good summary in Discussion. The empirical section was great as well.
Thoughts/Questions/Comments:
In general, the authors did a very good job using data and explaining just the minimal required about that data such that the reader can follow the figures displaying the profiles, the recurrence network, and the transition network. For Figure 2, although the figure does illustrate the point (why a stationary model would have missed the shifting structure regarding motivational variables), I think it will be natural for readers to try to make sense of what the shifting structure means over time. (Is this good/bad? Shift in usefulness of the separate items as students get bored answering them for the trillionth time? Some external factor (“history”) occurring that leads ultimately to simplified structure? Do these shifts just signal measures with poor reliability somehow?). That effort is likely to be frustrated because of the figure standing alone without a theoretical context. However, I can also see the drawbacks to flow if the authors have a long digression here in the text! (As a contrasting example, enough hints regarding context + interpretation—lines 470-472 + top of page 14—are offered to allow an interested reader to better understand the implications of Figure 5 and 6.)
- We have added the following text to the figure 2 caption: “Although this temporal variability can be due to e.g. changes in how the participant answers the questions (boredom, shifting perception of the items, etc.), or poor reliability of the measures, complexity theory would also guide us to expect that in very concrete reality, the direction and strength of relationships can shift over time and differ based on the state a person resides in. As an example, the relationships between motivational variables during behaviour change initiation phase, may differ from those during the maintenance phase.”
With regard to the limitations section: the authors seem to imply all measures in behavioral change research are momentary self-reported measures of psychological constructs. Given the wearables revolution, I would think that some of these limitations could be overcome if data from physiological, movement (activity), or recorders (E.A.R.) are theoretically informative and thus included as well.
- We added the following sentence to the end of the limitations section: “Another solution would naturally be tapping into wearable data; for example, electronically activated recorders [@kaplan2020BestPracticesElectronically] are maturing as a technology, and complexity methods have already been applied to physical activity data during a weight loss intervention [@chevance2020IdiographicDaytodayFluctuations].”
Over half of the data points in Figure 5 were either uncategorizable (about a third) or unique (nearly a fifth). That suggests quite a bit of data that is difficult to interpret. Given how bottom-up nonlinear dynamical analyses are, one concern I have long heard from researchers only familiar with conventional analyses is whether the analyses would find “something” even if the data were nonsense—that it will capitalize on noise or error, and find some kind of structure to fit the data from whatever one feeds into it. The challenge is how does a researcher validate that a structure that is found is different from what one would find by chance alone; isn’t this so post hoc rather than theory-driven? In some kinds of analyses (e.g., cross-recurrence quantification looking at how much one member of a dyad’s movement coordinates with their partner) there are analyses that can be done to give a sense of what results would result from nonsense data. For instance, in that example, one can do “pseudo-pair” analyses to show that if one mismatches the movement data from one member of a dyad with that of a member of a different dyad, the percent recurrence is substantially lower. I wonder if the authors might address the question of validating that the structures found in analyses mean anything or if SOME kind of structures would emerge even if there was no real “there” there (e.g., if the data were shuffled). At least some areas of psychology are famously skilled at coming up with a post hoc interpretation of whatever patterns appear in the data, after all.
- The problem of seeing something where there is nothing is an excellent observation, and one that deserves somewhat elaborate analysis. We have thus added a section (https://git.io/JqRTQ) to the supplementary website, outlining the results of surrogate data analysis.
- This analysis is briefly described in the manuscript after introducing transition networks: “To distinguish whether the results reflect non-linear structure in the data or are merely a product of randomness, the researcher can take advantage of a technique called surrogate data analysis [@schreiber2000SurrogateTimeSeries]. The analysis is presented in the supplementary website (section https://git.io/JqRTQ), but in brief, temporally disordered versions of the data—called “surrogates”—are created, and the observed data is compared to those. The surrogates represent the hypothesis that the data were generated by a rescaled Gaussian linear process. This means that, by analysing the surrogates, we ask whether the data can be understood to have arisen from a process, that is essentially stochastic and linear instead of highly interdependent and non-linear. The analysis indicates, that it would indeed be very unlikely to see these results, if the dynamics were Gaussian.”
- As regards the number of states uncategorizable or unique, this is dependent on the recurrence rate: A higher recurrence rate produces a larger radius, which allows for more leeway in how heterogeneous the states classified in each attractor can be. The appropriate recurrence rate is ultimately a research question of its own. We used an algorithm to find a radius parameter which produces 5% recurrence, but there are no hard and fast rules to make a universal choice. In the supplementary website we now present a robustness analysis for radius parameters leading to 5-10% recurrence rates – this shows that as the recurrence rate target increases, more patterns get classified outside the “unique” category. The patterns inside the “uncategorised” category could also be classified, but as they only consisted of just a few occurrences, we did not consider them of too much practical importance and hence binned them under “uncategorised”.
This question follows closely, to my mind, from the last point. I think one challenge for good nonlinear analyses (and interpretation) is that it seems to me that stronger theorizing might be required for successful, sophisticated nonlinear analyses; in contrast I feel one might be able to get by with less theoretical sophistication if one is relying on familiar regression analyses for instance. One worry I have about this excellent paper follows from how easily the authors convey a deep theoretical grasp not just of nonlinear complex systems but also theoretical bases of behavioral change interventions. But if someone uses nonlinear techniques, with little theoretical depth, the problems I mention in the previous comment might be particularly concerning. For me it comes through just how well thought out the theoretical side is for these authors (e.g., lines 488-495), but I don’t know if that will be apparent to a casual reader. Because the authors convey the theorizing side so smoothly that part of the work might look easy to a novice who would not know how much more imperative it is to do good theorizing in advance.
- We thank the reviewer for this praising comment, and agree theoretical considerations should be at the fore whenever possible. With that said, existing theories are borne almost entirely out of linear and construct-dominant paradigms that are not always compatible with a complexity-based approach. As applying complexity science to behavior change is a nascent area, it is difficult at this stage to identify exactly how complexity phenomena will impact upon existing theories, or indeed what new behavioral theories arising from complexity science findings might look like. We have included this sentence near the end of the discussion which hopefully makes this point: “information obtained from individual-level studies of dynamic complexity can then possibly inform models of larger groups, leading to better (or at least humbler and more nuanced) social scientific theories”
Typos:
Sentence in line 359-360: remove “to” after “provide”.
Line 326: in “collected by the participant” change “by”.
Line 536: remove extra “are”.
- These are now fixed, thank you for noticing. We have also also fixed other minor typos and language, mainly regarding usage of that/which. We express our gratitude for this thorough review.
Other changes:
- In the supplementary website, we had a figure depicting different centrality indices calculated from the transition network. As we do not introduce centrality indices, and their application to transition networks likely needs some additional work in order to be a more mathematically robust procedure, we have decided to omit the figure.
- In the supplementary website’s recurrence quantification section, we now also present robustness analysis for changing from Euclidean distance to Chebyshev distance.
- We discovered a small inconsistency in the approach to classifying the attractor states: The “1st” attractor was defined by node degree instead of strength centrality. While node degree and strength centrality correlate > 0.95 (see new section on surrogate data analysis on supplementary website), defining the 1st attractor by strength centrality instead of degree changes the results slightly.
Reviewer 3 Report
This paper provided an interesting, although dense, and somewhat unfocused read on the complexity science approach to behavior change. I very much agree with the authors that the science of behavior change must increasingly consider the interconnectedness of different behavioral predictors, the issue of differences of between-person differences and within-person change, and non-linear processes. These issues, however, are hardly radically new. Rather than dwelling on these issues, I therefore would find it more fruitful if the authors could shorten this part and expand more on “their” potential solution (complexity science), and compare this approach more in-depth to other approaches. I have listed further suggestions that may be considered when revising this manuscript.
The article discusses the issues of ergodicity, non-linearity etc. mostly from a methodological point of view. The manuscript might benefit from highlighting the theoretical implications of these issues. For example, as others have emphasized, what behavioral science is currently lacking is theorizing about the dynamics of behavior change (e.g. Scholz, 2019).
In the limitations section, the authors mostly dwell on the issues pertaining to collecting intensive longitudinal data. However, it would seem more to the point to discuss potential drawbacks of the complexity approach as that is what this paper focuses on.
In my opinion, science needs to strike a balance between simplification and complexity to provide useful understanding. Models are always an abstraction of reality and therefore to some extent incorrect. Hence, I think the paper would benefit from discussing the utility of taking the complexity approach to behavior change in relation to the goals the research aims to achieve. In behavior change, there can be different goals, particularly to explain, predict, and change behavior. It might be helpful to discuss the added benefit of the complexity (and potential drawbacks) of this approach (also compared to other approaches) for these different research goals.
In particular, the multilevel modelling approach seems to be an alternative way of handling the temporal dynamics, issues of ergodicity. How does this approach compare? The manuscript may benefit from elaborating more on this alternative approach (e.g. about mediation over time, Berli et al., 2020, or using spline models to model non-linear change processes, Inauen et al., 2017).
When discussing the issue of ergodicity (section 2.2) it seems to me that two issues were mixed that are separate: (1) Deriving conclusions about within-person change from between-person differences and (2) concluding from an average effect to a specific individual (heterogeneity).
On p. 10 the authors state “we cannot usually predict what the value of the next observation will be”. We can if we use simulation approaches. This is one way to handle complexity, and the paper may benefit from integrating this approach, particularly where researchers have done this in a theory- and evidence-based way (e.g. Tobias, 2009).
There seemed to be a disconnect between Table 1 and the corresponding text. Isn’t Table 1 supposed to summarize sections 2.1 to 2.3? It seemed to me that this could be more aligned. The authors should also ensure that all statements in Table 1 are sufficiently supported by the text / cited literature.
Figure 4: I didn’t understand this part of the caption: “Yellow nodes indicate the strongest state, red nodes the second strongest, followed by purple and blue.”. As stated earlier in the caption, the colors reflect “different motivation profiles”. So what is meant by the strength of the state?
References:
Berli, C., Inauen, J., Stadler, G., Scholz, U., & Shrout, P. E. (2020). Understanding between-person interventions with time-intensive longitudinal outcome data: Longitudinal mediation analyses. Annals of behavioral medicine, Epub-ahead.
Inauen, J., Bolger, N., Shrout, P. E., Stadler, G., Amrein, M., Rackow, P., & Scholz, U. (2017). Using smartphone‐based support groups to promote healthy eating in daily life: A randomised trial. Applied Psychology: Health and Well‐Being, 9(3), 303-323.
Scholz, U. (2019). It's time to think about time in health psychology. Applied Psychology: Health and Well‐Being, 11(2), 173-186.
Tobias R. Changing behavior by memory aids: a social psychological model of prospective memory and habit development tested with dynamic field data. Psychol Rev. 2009 Apr;116(2):408-38. doi: 10.1037/a0015512. PMID: 19348548.
Author Response
This paper provided an interesting, although dense, and somewhat unfocused read on the complexity science approach to behavior change. I very much agree with the authors that the science of behavior change must increasingly consider the interconnectedness of different behavioral predictors, the issue of differences of between-person differences and within-person change, and non-linear processes. These issues, however, are hardly radically new. Rather than dwelling on these issues, I therefore would find it more fruitful if the authors could shorten this part and expand more on “their” potential solution (complexity science), and compare this approach more in-depth to other approaches.
- Thank you for this comment. We acknowledge that this piece represents a general overview which does not provide any in-depth comparisons with other specific approaches. Rather, it acts as a bird’s eye comparison of several conventional approaches that form the current mainstream of behaviour change science, and ties these together with the rationale of idiographic science. We agree that our suggestions are not radically new, but the focus on idiographic methods within behavioral science is rather recent, and many unresolved and misunderstood issues remain in the literature. Furthermore, as per our experience based on conferences and discussions in the community, behavior change researchers are largely unfamiliar with the issues raised in this manuscript. Hence we feel that the presentation of these introductory elements is vital, despite them seeming self-evident to more advanced methodologists.
I have listed further suggestions that may be considered when revising this manuscript.
The article discusses the issues of ergodicity, non-linearity etc. mostly from a methodological point of view. The manuscript might benefit from highlighting the theoretical implications of these issues. For example, as others have emphasized, what behavioral science is currently lacking is theorizing about the dynamics of behavior change (e.g. Scholz, 2019).
- We agree that behavioral science has not yet advanced to theorizing about the dynamics of behavior change, and hope that we can one day achieve that. An important first step in this direction is to collect more within-persons data using intensive longitudinal methods, which allow for observing complex dynamics in the first place. Without such basic observational data as a starting point, it is difficult to create theories involving complex dynamics or to speculate on how complex dynamics might affect existing theories.
- We added a reference to the Scholz article after the sentence: “The role of time brings added complexity to this behavioural world, as dynamic patterns change over time and at varying frequencies.”
- We also added the following text to the discussion section, just before limitations:
- “Generating theory in this way would answer calls to address the issue of time more clearly in theories of health behaviour [@scholz2019ItTimeThink]. It could also lay the foundation for more formal theories of behaviour change to be developed [@chevance2020InnovativeMethodsPredicting], as these typically hypothesise how relationships between variables unfold over time, and a more coherent correspondence between theoretical cycles and empirical cycles in behaviour change research [@vanrooij2021TheoryTestHow].”
In the limitations section, the authors mostly dwell on the issues pertaining to collecting intensive longitudinal data. However, it would seem more to the point to discuss potential drawbacks of the complexity approach as that is what this paper focuses on. In my opinion, science needs to strike a balance between simplification and complexity to provide useful understanding. Models are always an abstraction of reality and therefore to some extent incorrect.
Hence, I think the paper would benefit from discussing the utility of taking the complexity approach to behavior change in relation to the goals the research aims to achieve. In behavior change, there can be different goals, particularly to explain, predict, and change behavior. It might be helpful to discuss the added benefit of the complexity (and potential drawbacks) of this approach (also compared to other approaches) for these different research goals.
- We appreciate the point. In this manuscript, we have opted to mainly focus on the change aspect of behaviour change, and to keep the piece from reaching excessive lengths, feel this is a justifiable decision. On the other hand, we do mention and refer the reader to Aaron Fisher’s work on predicting smoking with idiographic models, acknowledge how coevolution leads to prediction problems, discuss the relationship between prediction and non-stationarity, and introduce research where behavioural phase transitions are predicted by critical fluctuations. That latter line of research is an example of what can be aspired to to; an understanding of first principles (i.e. explanation) leads to either testable predictions of what changes behaviour under which conditions. In a sense, the three goals mentioned by the reviewer are not separate but parts of a larger whole. Complexity science implies changes to our conventional perspective of billiard ball type causes and effects – many of the implications, such as the hard limits to prediction horizons set by sensitivity to initial conditions did not fit the scope of this manuscript and are expressed in other work currently under progress by members of the author group. Being in a way knowledge about the limits of to knowledge (Allen, 2001) – and as the necessity of the complexity approach can be evaluated with tests presented in this paper and e.g. Olthof (2020) – the drawbacks apart from practical matters of data collection seem to mostly concern problems of sociology of science: Although the complexity approach implies that many conventional practices are no longer sound, early adopters applying the perspective may encounter considerable barriers to publishing their results, which rely on foreign-sounding concepts and give less definitive answers regarding more opaque causal processes. We view these issues to lie outside the scope of the paper.
In particular, the multilevel modelling approach seems to be an alternative way of handling the temporal dynamics, issues of ergodicity. How does this approach compare? The manuscript may benefit from elaborating more on this alternative approach (e.g. about mediation over time, Berli et al., 2020, or using spline models to model non-linear change processes, Inauen et al., 2017).
- We tend to agree with what e.g. Piccirillo and Rodebaudh (2019) consider to be the main issue with multilevel models: “Notably, in multilevel models, within-person variability is pooled across individuals, rather than person-specific (Bringmann et al., 2013). Therefore, some researchers would question the inclusion of multilevel models in this review at all: These models only handle individuals as departures from group-level means. There may be instances in which such a model is completely appropriate, but multilevel models in isolation cannot provide evidence as to whether this is the case or not. For example, multilevel models must make an assumption about how the group’s parameters are distributed, and this assumption might be incorrect. If the assumption is incorrect, the individual parameter estimates will also be incorrect. In the absence of prior information regarding the distributions, consulting fully idiographic methods in combination with a multilevel model would be necessary to determine whether the multilevel model was reasonably appropriate.”
- That said, e.g. the person-specific multilevel survival model used in the early warning signal studies cited in the paper can be perfectly sound.
- To make the point clearer, we have changed the sentence “This enables us to observe more granularity in the dynamics, than allowed by e.g. multilevel models” to the following: This enables us to observe more granularity in the dynamics, than allowed by e.g. multilevel models, which treat individuals as departing from group-level means according to a known distribution [@piccirillo2019FoundationsIdiographicMethods].
When discussing the issue of ergodicity (section 2.2) it seems to me that two issues were mixed that are separate: (1) Deriving conclusions about within-person change from between-person differences and (2) concluding from an average effect to a specific individual (heterogeneity).
- Ergodicity describes the conditions under which the analysis of intra-individual variation would yield the same results as an analysis of inter-individual variation, and hence the two issues are tied together. The point is formally described in Molenaar (2008) in relation to (classical) test theory, but we consider this too technical an analysis for the general audience. To more simply convey our points about non-ergodicity, we have rewritten the first four paragraphs of that section. These now read as follows:
To be useful to individuals, processes postulated by psychology ought to work on the individual level [@johnstonUsefulTheoriesShould2013]. Whether group-level variation is informative of individual-level dynamics, depends on a condition known as ergodicity, which has the following properties: "Only if the ensemble of time-dependent trajectories in behavior space obeys two rigorous conditions will an analysis of interindividual variation yield the same results as an analysis of intraindividual variation [...] First, the trajectory of each subject in the ensemble has to obey exactly the same dynamical laws (homogeneity of the ensemble). Second, each trajectory should have constant statistical characteristics in time (stationarity, i.e., constant mean level and serial dependencies)" ([@molenaar2008ConsequencesErgodicTheorems]; see also [@molenaarImplicationsClassicalErgodic2008]).
In other words, this would mean that in a 100x100 spreadsheet, where participants are rows and measurement occasions are columns, calculating an average of values within one column ("ensemble average"), would give the same result as calculating the same statistic from one row ("time average"). For example, in an ergodic process, the mean and standard deviation of each person’s daily minutes of physical activity over a 100-day period would be the same as the mean and standard deviation of 100 people’s daily physical activity minutes measured once. Or, observing that 20% of a given population are smokers, would mean that everyone is a smoker for 20% of their lives. In terms of coupled processes, the correlation between physical activity and intention would be the same in the population measured once, as it is for one person over time.
Going back to the two "rigorous conditions", the condition of homogeneity almost by definition rules out the behaviour change researcher’s interests, as we are interested in how people (can) change, and it is quite clear that people do not all follow the same behaviour change processes. Indeed, it would seem preposterous to suggest that, for example, self-regulation is a constant process during a individual’s life span. Although the mathematical proof for the non-equivalence of inter-individual and intra-individual data structures was published over a decade ago [@molenaarManifestoPsychologyIdiographic2004], only recently has serious research attempted to quantify the threat stemming from lack of group-to-individual generalisability [@fisherLackGrouptoindividualGeneralizability2018]. This preliminary work indicates that even if we could work with "generalisable" ideal random samples from well-defined populations, we would still be committing the ecological fallacy (i.e. drawing individual-level inferences from group-level data) if we wanted to apply our knowledge to individuals.
The second condition, that the statistical properties of these processes must not change over time, is generally referred to as stationarity. In the context of physical activity, the extent to which physical activity is influenced by other factors, is likely to change over time. For example, the effect of discomfort on physical activity is likely to change in a non-linear manner over time, as fitness and tolerance of discomfort fluctuate not only because of randomness, but as core features of the phenomena itself [@hallTemporalSelfregulationTheory2015]. However, the tools most often used in research for thinking about and analysing behaviour change, such as linear regression, do not account for these kinds of temporal dynamics. This is because temporal cognitive change fundamentally violates the assumption of stationarity, as exemplified next.
On p. 10 the authors state “we cannot usually predict what the value of the next observation will be”. We can if we use simulation approaches. This is one way to handle complexity, and the paper may benefit from integrating this approach, particularly where researchers have done this in a theory- and evidence-based way (e.g. Tobias, 2009).
- What the statement in context was meant to convey is, as Tobias (2009) puts it, that a model can approximately “replicate the global tendencies of the dynamics”, without being precisely correct about any specific instances. While research lines stemming from Sugihara et al. (2012)—recently emerging as empirical dynamic modeling (Sugihara et al., 2020)—and the early warning signal studies discussed in our manuscript show some promise, exact prediction beyond some short-term horizon is convincingly shown to be impossible in complex and not-so-complex systems across sciences (as indicated since Poincare’s discovery of the famous three-body problem). We do believe that simulation approaches can fruitfully be used to evaluate how difficult prediction is going to be, and how long the aforementioned prediction time horizon is – as discussed in Olthof (2020). On the other hand, this otherwise very deserving topic seems to fall outside of the scope of the current manuscript.
There seemed to be a disconnect between Table 1 and the corresponding text. Isn’t Table 1 supposed to summarize sections 2.1 to 2.3? It seemed to me that this could be more aligned. The authors should also ensure that all statements in Table 1 are sufficiently supported by the text / cited literature.
- Thank you for this comment. We have adjusted the text that introduces Table 1. Each of the 3 key terms (interconnectedness, non-ergodicity, and non-linear dynamics) are now present, thereby making the link between the text and the table more 1:1. We have also added several references to important existing literature within the table.
Figure 4: I didn’t understand this part of the caption: “Yellow nodes indicate the strongest state, red nodes the second strongest, followed by purple and blue.”. As stated earlier in the caption, the colors reflect “different motivation profiles”. So what is meant by the strength of the state?
- As expressed in the supplementary website, our approach to classifying the states capitalises on the strength centrality of the nodes in the recurrence network. To clarify this, the figure caption now reads: “Weighted multidimensional recurrence network. Each circle ("node") is a measurement occasion, numbers indicate their running number, and colors represent different motivation profiles. These profiles are configurations of six variables, and can be conceived of as attractors. Lines indicate the same motivational state reoccurring at a later time point. Yellow nodes indicate configurations connecting to that with the highest strength centrality (i.e. number of connections weighted by the similarity of the connected nodes), red nodes connect to the second strongest which is not connected to the strongest, followed by purple and blue. Grey nodes depict uncategorised configurations which occur at least twice, and white ones the configurations, which only occur once. Nodes that are larger, are connected to more other nodes. Drawn with R package casnet [@hasselman2020CasnetToolboxStudying].”
References:
Berli, C., Inauen, J., Stadler, G., Scholz, U., & Shrout, P. E. (2020). Understanding between-person interventions with time-intensive longitudinal outcome data: Longitudinal mediation analyses. Annals of behavioral medicine, Epub-ahead.
Inauen, J., Bolger, N., Shrout, P. E., Stadler, G., Amrein, M., Rackow, P., & Scholz, U. (2017). Using smartphone‐based support groups to promote healthy eating in daily life: A randomised trial. Applied Psychology: Health and Well‐Being, 9(3), 303-323.
Scholz, U. (2019). It's time to think about time in health psychology. Applied Psychology: Health and Well‐Being, 11(2), 173-186.
Tobias R. Changing behavior by memory aids: a social psychological model of prospective memory and habit development tested with dynamic field data. Psychol Rev. 2009 Apr;116(2):408-38. doi: 10.1037/a0015512. PMID: 19348548.
Other changes:
- In the supplementary website, we had a figure depicting different centrality indices calculated from the transition network. As we do not introduce centrality indices, and their application to transition networks likely needs some additional work in order to be a more mathematically robust procedure, we have decided to omit the figure.
- In the supplementary website’s recurrence quantification section, we now also present robustness analysis for changing from Euclidean distance to Chebyshev distance.
- We discovered a small inconsistency in the approach to classifying the attractor states: The “1st” attractor was defined by node degree instead of strength centrality. While node degree and strength centrality correlate > 0.95 (see new section on surrogate data analysis on supplementary website), defining the 1st attractor by strength centrality instead of degree changes the results slightly.